# Inflammatory Indices vs. CA 125 for the Diagnosis of Early Ovarian Cancer: Evidence from a Multicenter Prospective Italian Cohort

**DOI:** 10.3390/jpm15090426

**Published:** 2025-09-04

**Authors:** Carlo Ronsini, Stefano Restaino, Manuela Ludovisi, Giuseppe Vizzielli, Mariano Catello Di Donna, Giuseppe Cucinella, Maria Cristina Solazzo, Cono Scaffa, Pasquale De Franciscis, Mario Fordellone, Stefano Cianci, Vito Chiantera

**Affiliations:** 1Unit of Gynecologic Oncology, National Cancer Institute, IRCCS, Fondazione “G. Pascale”, 80131 Naples, Italy; mariano.didonna@istitutotumori.na.it (M.C.D.D.); giuseppe.cucinella@istitutotumori.na.it (G.C.); mariacristinasolazzo@stuedenti.unicampania.it (M.C.S.); c.scaffa@istitutotumori.na.it (C.S.); vito.chiantera@istitutotumori.na.it (V.C.); 2Unit of Obstetrics and Gynecology, “Santa Maria della Misericordia” University Hospital, Azienda Sanitaria Universitaria Friuli Centrale, 33100 Udine, Italy; stefano.restaino@policlinicogemelli.it (S.R.); giuseppe.vizzielli@yahoo.it (G.V.); 3Department of Life, Health and Environmental Sciences, University of L’Aquila, 67100 L’Aquila, Italy; 4Unit og Gynaecology and Obstetrics, Department of Woman, Child and General and Specialized Surgery, University of Campania “Luigi Vanvitelli”, 80131 Naples, Italy; pasquale.defranciscis@unicampania.it; 5Medical Statistics Unit, Department of Mental and Physical Health and Preventive Medicine, University of Campania “Luigi Vanvitelli”, 80131 Naples, Italy; mario.fordellone@unicampania.it; 6Unit of Gynecology and Obstetrics, Department of Human Pathology of Adult and Childhood “G. Barresi”, University of Messina, 98125 Messina, Italy; stefano85@hotmail.it

**Keywords:** ovarian neoplasm, borderline ovarian tumor, systemic inflammatory response syndrome, CA 125 antigen, biomarkers

## Abstract

Ovarian cancer (OC) diagnosis remains challenging due to the low specificity of CA 125, requiring additional biomarkers for improved accuracy. This prospective multicenter study analyzed 94 patients with adnexal masses, including 31 benign tumors, 42 borderline ovarian tumors (BOTs), and 21 OC cases. We assessed the diagnostic performance of the Systemic Inflammation Response Index (SIRI) and the Systemic Inflammatory Response (SIR) compared to CA 125. Our results demonstrated that SIRI had superior diagnostic accuracy (AUC = 0.71) compared to CA 125 (AUC = 0.59). Regression analysis confirmed SIRI as an independent predictor of non-benign ovarian tumors (*p* = 0.01). These findings suggest systemic inflammatory indices could enhance risk stratification and early OC detection, offering a cost-effective and accessible alternative to traditional biomarkers. Further research is needed to validate and integrate these results into clinical practice.

## 1. Introduction

Ovarian cancer (OC) remains one of the most challenging gynecologic malignancies to diagnose in its early stages, significantly impacting prognosis and treatment options [1]. The lack of reliable early diagnostic tools often leads to delayed intervention, reducing the chances of curative treatment and increasing the need for aggressive therapeutic strategies [2,3]. Distinguishing between benign ovarian masses, borderline ovarian tumors (BOTs), and malignant neoplasms is essential for optimizing patient management and avoiding unnecessary invasive procedures [4].

Several diagnostic models have been developed to enhance preoperative evaluation [5,6,7,8,9], including the Assessment of Different Neoplasias in the Adnexa (ADNEX), formulated by the International Ovarian Tumor Analysis (IOTA) group [10]. While ADNEX has shown high sensitivity in differentiating benign from malignant lesions, its accuracy declines when distinguishing between benign formations, early-stage OC, and BOTs. A similar attempt, with similar results, also arises from MRI analysis [11]. This distinction is particularly relevant as BOTs and OC require different surgical and therapeutic approaches [4], and late diagnosis can severely affect patients’ prognosis [12].

Among the biomarkers explored in ovarian cancer diagnostics, CA 125 remains the most widely used [13,14,15]. However, its limited specificity reduces its clinical utility [16], as elevated levels are also observed in benign conditions such as endometriosis, pelvic inflammatory disease, and other inflammatory and hepatic disorders [17,18,19,20]. To refine diagnostic accuracy, novel biomarkers are needed, particularly those reflecting the complex interplay between cancer progression and the immune response.

Systemic inflammatory indices, such as the Systemic Inflammation Response Index (SIRI) and the Systemic Inflammatory Response (SIR), have shown prognostic and diagnostic value in various malignancies [21,22,23]. These indices are particularly promising because they can be obtained at no additional cost as they are derived from routine blood tests already performed in standard preoperative assessments [24]. In addition, they can potentially reflect the individual’s response to tumoral progression. It is likely that upon the tumor tissue’s acquisition of infiltrative capacity, it comes in contact with the patient’s immune system, resulting in its response, which can potentially be measured. This makes them an accessible and cost-effective tool to complement existing diagnostic methods. If validated, these inflammatory indices could serve as an additional tool in the preoperative monitoring of suspicious adnexal masses, improving risk stratification and potentially reducing the number of unnecessary surgeries. Currently, many patients with benign lesions undergo surgery due to diagnostic uncertainty [25]. By integrating SIRI and SIR into clinical decision-making, it may be possible to better identify patients who require immediate surgical intervention and those who can be safely monitored over time, thereby reducing the burden on surgical teams and minimizing patient exposure to operative risks.

### 1.1. Objective

This study aims to evaluate whether inflammatory indices (SIRI and SIR) are elevated in patients with ovarian cancer (OC) and borderline ovarian tumors (BOTs), and whether their combination improves diagnostic performance compared to CA 125 alone. To explore this, we conducted a prospective observational multicenter cohort study, enrolling patients with suspected adnexal masses without evidence of extra-ovarian disease. We compared the mean value of inflammatory indices and CA 125 in benign lesions, OC, and BOTs.

### 1.2. Secondary Objectives

Additionally, we aim to evaluate their diagnostic power by constructing ROC curves and estimating the optimal cut-off values to improve differential diagnosis and refine patient selection for surgery.

## 2. Material and Methods

### 2.1. Study Design

We conducted a multicenter prospective observational cohort study utilizing secondary data from a broader clinical registry related to the multicentric study “Ovarian Cyst Enucleation Spillage Score” (NTC05376384). This study focused on patients diagnosed with adnexal masses who underwent primary surgical staging at three specialized gynecologic oncology units: the University of Campania Luigi Vanvitelli in Naples, the Santa Maria della Misericordia University Hospital in Udine, and the University of Messina in Italy.

This study adhered to the STROBE guidelines for observational research [26]. To ensure compliance with ethical and privacy regulations, specific informed consent forms for data processing were required from all participants. According to local regulatory frameworks, this study was approved by the Ethical Committee of the University of Campania Luigi Vanvitelli (Approval No. 0013958/I 5 May 2022). This research was aligned with the principles outlined in the Helsinki Declaration.

The primary aim of this study was to evaluate differences in systemic inflammatory indices (SIR and SIRI) and CA 125 among patients with benign adnexal tumors, BOTs, and OC. To assess the diagnostic potential of these biomarkers, we performed ROC curves analysis and constructed a linear regression model to explore their association with BOT and OC diagnoses.

### 2.2. Setting

Between January 2023 and January 2025, we prospectively collected data from patients treated for ovarian cysts at three participating centers. Those with a histological diagnosis of OC or BOTs underwent staging surgery, which included hysterectomy, bilateral adnexectomy, omentectomy, peritoneal biopsies, and, when OC was diagnosed, lumboaortic and pelvic lymphadenectomy to confirm early-stage disease. Otherwise, the following were excluded from this study.

Preoperative blood samples were collected within seven days before surgery, including a quantitative white blood cell assay and its characterization. Additionally, serum CA 125 levels were measured within 15 days preoperatively. Based on histological findings, patients were classified as benign, BOT, or OC. Two independent expert pathologists reviewed and confirmed the final diagnosis.

### 2.3. Participants

Patients were eligible for inclusion if they had a unilateral adnexal mass identified via transvaginal pelvic ultrasound within 30 days before surgery. Additional requirements included a complete blood count performed within 7 days before surgery, a serum CA 125 measurement within 15 days preoperatively, and the completion of surgical staging with a confirmed histological diagnosis. The ADNEX model was used for risk stratification, with patients included only if their probability of disease beyond FIGO Stage I [27] was ≤10%. Furthermore, participants needed to have comprehensive clinical status documentation based on either a preoperative CT scan or a total-body PET scan, showing no evidence of extra-ovarian disease in the case of a diagnosed BOT or OC. Imaging studies must have been conducted within 30 days after surgery and reviewed independently by two blinded radiologists. The ADNEX model [10] was used for risk stratification, with patients included only if their probability of disease beyond FIGO Stage I was ≤10%.

Exclusion criteria included chronic systemic inflammatory conditions such as Crohn’s disease, ulcerative colitis, systemic lupus erythematosus, multiple sclerosis, Hashimoto’s thyroiditis, non-alcoholic fatty liver disease, fibromyalgia, chronic kidney disease, hepatitis, osteoarthritis, or psoriasis. Additionally, patients were excluded if they had a histological diagnosis of ovarian or extra-ovarian endometriosis. Those with a history of other malignancies diagnosed within the last three years, conditions causing excessive corticosteroid production, or who had received steroid therapy in the 30 days preceding blood collection were also not eligible. Finally, patients with incomplete clinical data or an uncertain histological diagnosis of OC or BOT were categorized under an “intention-to-treat” classification, as illustrated in the flowchart (Figure 1).

### 2.4. Variables

This study analyzed several variables, including Body Mass Index (BMI) (kg/m^2^) and age (years), both considered as continuous variables. The histological classification of ovarian neoplasia was treated as an ordinal variable, divided into three categories: benign, BOT, and OC, with further stratification based on specific histotype, as independently assessed by two pathologists.

Laboratory parameters included neutrophils, monocytes, lymphocytes, and platelets, which were quantified as 10^3^ units/dL and considered continuous variables. Serum CA-125 levels were also measured in IU/dL and treated as continuous variables.

To assess the inflammatory response, two derived indices were calculated: the Systemic Inflammatory Response (SIR), obtained by multiplying the neutrophil count by the platelet count and dividing by the lymphocyte count, and the Systemic Inflammatory Response Index (SIRI), determined by multiplying monocytes by platelets and dividing by lymphocytes. Both were considered continuous variables and were used as the principal outcome of differentiation between histological categories.

### 2.5. Laboratory

Peripheral blood samples (3.0 mL) were collected from the ulnar vein within seven days before surgery for hematological analysis. To prevent coagulation, samples were drawn using a sterile vacuum collection system and immediately placed in tubes containing ethylenediaminetetraacetic acid (EDTA). After gentle inversion to ensure proper anticoagulation, samples were stored at 4 °C until processing. Within two hours of collection, an automated hematology analyzer was used to measure neutrophils, lymphocytes, monocytes, eosinophils, basophils, and platelet counts, with results reported in 10^3^ units/dL. For CA 125 quantification, additional blood samples were collected within 15 days before surgery. These were drawn into serum separator tubes (SSTs), allowed to clot at room temperature for 30 min, and then centrifuged at 3000× *g* for 10 min to isolate the serum. The separated serum was then aliquoted and stored at −80 °C until further analysis. Measurements were conducted using an electrochemiluminescence immunoassay (ECLIA) on a Cobas e411 analyzer (Roche Diagnostics, Basel, Switzerland), ensuring high sensitivity and specificity in detecting CA 125 levels.

All participating centers adhered to standardized pre-analytical and analytical procedures for blood sample collection and processing. Complete blood counts and CA 125 levels were analyzed using automated, calibrated instruments, each undergoing daily internal quality checks. Additionally, all laboratories participated in national external quality assessment (EQA) programs to ensure inter-laboratory consistency and diagnostic accuracy. Staff at each site were trained to follow harmonized protocols for sample handling, timing, and storage to minimize variability across centers.

All laboratory analyses were performed on-site at the three participating centers, adhering to standardized protocols to maintain accuracy and reproducibility.

### 2.6. Statistical Analysis

The normality of continuous variables was assessed using the Kolmogorov–Smirnov test. Categorical variables were summarized as absolute frequencies and percentages, with comparisons between groups performed using Fisher’s exact and Chi-square tests. Continuous variables were presented as medians with interquartile ranges (Q1–Q3) and analyzed using the Wilcoxon test for two-group comparisons. In contrast, the Kruskal–Wallis test compared more than two independent groups.

Based on their histological diagnosis, patients were categorized into benign, BOT, and OC. The null hypothesis (H_0_) assumed no significant differences in mean SIR and SIRI values among the three groups (H_0_: µ_1_ = µ_2_ = µ_3_), while the alternative hypothesis (H_1_) suggested a significant variation (H_1_: µ_1_ ≠ µ_2_ ≠ µ_3_, two-sided test).

The sample size calculation aimed to detect a minimum difference of 0.76 standard deviations in the mean SIR and SIRI values among the three groups. Due to the lack of prior data, this estimate was based on the assumption of a notable distinction between groups, aligning with biological evidence from other malignancies. To achieve 80% statistical power at a significance level of α = 0.05, a minimum of 21 patients per group was required. The sample size calculation was performed using R (pwr package).

A multivariate linear regression model was applied to assess the association between inflammation indices and relevant clinical parameters. The model’s significance was evaluated using the maximum likelihood method.

ROC curve analysis was conducted for SIR, SIRI, and CA 125 to assess diagnostic performance, differentiating benign from non-benign (BOTs or OC) cases. The area under the curve (AUC) was calculated to determine diagnostic accuracy, applying the 2000 bootstrap resampling method. The Youden Index was used to establish optimal cutoff values for SIR and SIRI, while the CA 125 cutoff was based on clinical standards and set at 39 IU/dL (<40 IU/dL) [28].

Boxplots were generated to visualize the distribution of continuous variables across outcome groups. All statistical analyses were conducted using R software (version 2024.12.0 + 467) and RStudio (version 2024.12.0 + 467). ROC curves and AUC values were derived using the ROC package, with a *p*-value < 0.05 considered statistically significant.

### 2.7. Risk of Bias

Multivariate regression analyses were performed to reduce potential confounders, incorporating all available variables. The resulting models were assessed using adjusted R^2^ and Bayesian Information Criterion (BIC), with the best-fitting model selected based on the lowest BIC value.

CR initially conducted the data analysis, followed by an independent blinded review by MF, who was unaware of this study’s objectives. No missing data were identified in the key outcome variables.

### 2.8. Handling of Missing Data and Sensitivity Analysis

Missing data were addressed using multiple imputation techniques, ensuring the preservation of statistical power and reducing potential bias. The imputed values were cross-validated using complete case analysis to verify consistency. Sensitivity analyses involved excluding extreme values, testing alternative statistical models, and assessing subgroup-specific trends.

### 2.9. Declaration of Generative A.I. in Scientific Writing

The authors declare that no A.I. was used to write the original draft. Grammar correction tools (Grammarly, Inc., San Francisco, CA, USA), were used to improve the quality of English and readability. The technology was used under human oversight and control.

## 3. Results

Between January 2023 and January 2025, 174 patients were screened for inclusion in this study. Of these, 38 were excluded before surgery due to the presence of one or more exclusion criteria, and 136 underwent surgery. Due to the presence of post-surgical exclusion criteria, 42 patients were excluded from this study. A total of 94 patients were in the end enrolled: 31 benign, 42 BOT, and 21 OC, reaching the calculated minimum sample size of 21 patients for the single arm. All exclusion reasons are shown in Figure 1.

There was no statistically significant difference between the three groups regarding age, BMI, or menopausal status. However, patients in the OC group exhibited higher mean levels of CA 125, neutrophils, monocytes, and platelets. Table 1 provides a detailed summary of the population characteristics.

### 3.1. Outcomes

The primary outcome was to estimate the difference in mean value of SIR and SIRI between benign, BOT, and OC. SIR and SIRI were higher in the OC group (SIR 1347 vs. 699 vs. 557, *p* = 0.016; SIRI 2.73 vs. 1.36 vs. 0.76 *p* < 0.001). These results are shown in Table 2.

The distribution of the SIR, SIRI, and CA 125 values is graphed as a boxplot in Figure 2.

### 3.2. Linear Regression

A linear regression model was constructed to evaluate the relationship between non-benign ovarian diagnosis and SIR, SIRI, and CA 125 as independent variables. The analysis identified a statistically significant association between non-benign ovarian diagnosis (BOT or OC) and SIRI (beta coefficient of 1.2 95% CI: 0.3–2.1, *p* = 0.01). At the same time, SIR and CA 125 failed to reach a statistically significant association. Those data are shown in Table 3.

### 3.3. ROC Curve

We constructed ROC specifications for SIR, SIRI, and CA 125 to compare the potential diagnostic of non-benign adnexal neoplasia (Figure 3).

SIRI demonstrated the highest AUC (0.71, CI 95% 0.59–0.81) and CA 125 the lowest AUC (0.59, CI 95% 0.46–0.70). SIR showed intermediate performance (AUC 0.60, CI 95% 0.48–0.72). Using Youden’s method, the optimal diagnostic cut-off values were determined as 710 for SIR and 1 for SIRI, while for CA 125, the commonly used positivity threshold of 40 IU/dL was applied [28]. SIRI achieved the highest specificity (0.74) and sensitivity (0.70). Both the inflammatory indices outperformed CA 125 in sensitivity, positive predictive value (PPV), and negative predictive value (NPV). SIR, on the other hand, demonstrated specificity (0.68) equal to that of CA 125. All detailed cut-off performance metrics are provided in Table 4.

## 4. Discussion

### 4.1. Interpretation of Results

Our study demonstrates that SIR and SIRI levels progressively increase along the spectrum from benign neoplasms to BOTs and OC. This trend is primarily driven by a rise in neutrophils, monocytes, and platelets, while lymphocyte levels remain stable. From a biological perspective, we hypothesize that elevated inflammatory markers reflect a pro-inflammatory tumor microenvironment, which is known to promote tumor progression [29]. Additionally, tumor metabolism, driven by hypoxia and oxidative stress, may further modulate the host immune response [30]. Notably, our study exclusively included patients with ovarian-confined disease, minimizing confounding factors related to metastatic dissemination. This strengthens the hypothesis that the observed inflammatory response is an intrinsic characteristic of tumor biology rather than a secondary effect of systemic disease spread. Therefore, the mechanisms underlying malignant transformation appear to drive distinct immune responses in benign neoplasms, BOTs, and OC. In line with this observation, SIRI was also shown to differ significantly between benign neoformations and BOTs. Given their potential role in the early diagnosis of ovarian neoplasms, we conducted a linear regression analysis to confirm the association between systemic inflammation indices and non-benign adnexal masses, encompassing both BOTs and OC. The need to accurately distinguish benign from non-benign adnexal masses stems from the different clinical management these conditions provide and the different prognostic impacts [4,12]. Interestingly, our findings revealed that SIRI was the only marker to exhibit a statistically significant association in differentiating benign from non-benign formations. Surprisingly, CA 125, the most widely used preoperative biomarker for ovarian neoplasms, did not show a similar correlation. Consistently, ROC curve analysis also identified SIRI as the best-performing index, while both inflammation-based indices (SIR and SIRI) outperformed CA 125, although the diagnostic performance levels shown remain perfectible. Furthermore, the optimal cut-off values derived for these markers demonstrated higher positive predictive value (PPV) and specificity than the conventional 39 IU/dL threshold of CA 125, which remains the most widely used positivity cut-off for this tumor marker.

### 4.2. Clinical Implication

The SIR and SIRI inflammatory indices have demonstrated greater diagnostic accuracy than CA 125 in distinguishing benign from non-benign adnexal formations (BOTs and OC). This finding suggests that these indices could serve as valuable diagnostic tools for gynecologic oncologists. Interestingly, CA 125 remains the most commonly requested biomarker for adnexal masses despite its questionable diagnostic performance in this field [31]. Moreover, its widespread use burdens healthcare systems financially [32]. In the United States, a CA 125 assay costs between USD 20 and USD 105, with an average price of USD 68.

In contrast, SIR and SIRI can be derived from a standard complete blood count (CBC), an inexpensive test already included in preoperative evaluations. This makes them highly accessible and practical for clinical implementation, potentially offering a more cost-effective approach to risk stratification in patients with adnexal masses. Other groups of scholars have shown that this method is largely reproducible and cost-effective. Preoperative NLR combined with CA 125 has been shown to be a simple and cost-effective approach for identifying ovarian cancer [33]. Similarly, platelet to lymphocyte ratio (PLR) is considered an accessible and low cost biomarker for diagnosis and staging, especially when used adjunctively with other diagnostic tools [34].

Beyond cost-effectiveness, these inflammatory indices provide broader clinical information than CA 125. Recent evidence has further supported this approach. In particular, Opławski et al. demonstrated that systemic inflammatory biomarkers, alongside molecular signatures, are associated with treatment response and prognosis in ovarian cancer, highlighting their relevance in biomarker-driven clinical decision-making [35].

It is important to note that this study was conducted under highly stringent conditions, excluding patients with pre-existing pro-inflammatory conditions to avoid potential confounding factors. Consequently, 46% of “intention-to-treat” patients were excluded for failing to meet inclusion criteria, which may limit the real-world applicability of these indices.

Furthermore, while the absolute diagnostic performance of these indices is not exceptional, with the highest AUC reaching only 70%, the primary objective of this study was to demonstrate that inflammatory indices increase as ovarian neoplasms progress, which was clearly confirmed.

From a clinical perspective, an important observation is the poor performance of CA 125, a biomarker that is frequently overused in daily practice despite its limited diagnostic reliability. Although CA 125 is often integrated with imaging techniques to develop diagnostic models, similar applications could extend to inflammatory indices if validated by further research.

Ultimately, SIR and, more notably, SIRI could serve as valuable complementary diagnostic tools for monitoring suspicious adnexal formations and improving the early detection of malignant transformation.

### 4.3. Comparison with Existing Literature

The interplay between systemic inflammation and tumor progression has been widely studied across various malignancies, with increasing evidence supporting the role of inflammatory biomarkers as potential diagnostic and prognostic tools [21,22,23]. Like many other solid tumors, OC triggers a complex immune response that can be reflected in systemic inflammatory indices such as SIRI and SIR. Our findings demonstrate a progressive elevation of these indices from benign to malignant ovarian lesions, which are consistent with broader oncological research highlighting the link between inflammation and tumor biology [36].

Previous studies have shown that systemic inflammatory markers are already elevated at the earliest stages of tumor development, likely due to the tumor’s ability to manipulate the host immune system [34]. This is particularly relevant in gynecologic malignancies. In endometrial cancer, for instance, our research group previously identified a similar pattern, where an increase in inflammatory indices accompanied the transition from atypical endometrial hyperplasia to carcinoma [37,38]. This aligns with the current understanding, suggesting that inflammation is both an early warning signal and a key player in tumor progression.

Recent research has further refined our understanding of the immune landscape in OC. Blanc-Durand et al. have demonstrated that in the initial phases of ovarian cancer, a robust immune response is mediated by natural killer (NK) cells and tumor-infiltrating lymphocytes [39]. However, as the disease progresses, the chronicity of inflammation fosters an immunosuppressive environment, facilitating tumor angiogenesis, metastasis, and immune evasion. This shift is thought to be mediated by increased production of cytokines such as IL-6 and TGF-β1, which are also implicated in the modulation of extracellular vesicles in OC [40]. These findings underscore the dynamic nature of the immune response in ovarian malignancies and provide a biological rationale for the observed variations in SIRI and SIR levels in our study.

The significance of systemic inflammation in BOTs and OC extends beyond diagnosis to prognosis. Several studies have demonstrated that elevated inflammatory indices correlate with poorer overall survival and shorter platinum-free intervals in OC patients [41,42]. While our study focused primarily on the diagnostic implications of SIRI and SIR, this broader body of evidence suggests that these indices could also serve as valuable prognostic markers. Systemic inflammation’s ability to predict malignancy and stratify patient risk highlights its potential as a multifaceted tool in oncologic decision-making.

Our results also contribute to the ongoing discussion about the limitations of CA 125 as a standalone biomarker for ovarian tumor diagnostics. Despite being the most widely used serum marker, CA 125 has well-documented limitations, particularly in differentiating malignant from benign ovarian conditions [43]. Our findings reinforce this notion, as CA 125 exhibited lower sensitivity and specificity than SIRI, suggesting that inflammation-based indices may provide additional discriminatory power when incorporated into diagnostic models.

From a clinical standpoint, these insights emphasize the need for a more integrated approach to ovarian tumor diagnostics.

In conclusion, our findings align with and expand upon the existing literature demonstrating the diagnostic and potential prognostic utility of systemic inflammatory markers in ovarian masses. By reaffirming the link between inflammation and malignancy, our study supports the growing recognition of inflammatory indices as promising tools for early detection and risk stratification in ovarian tumors.

### 4.4. Strengths and Limitations

Our study finds its strength in the strong statistical significance of the results, methodological rigor, and its prospective nature. In addition, this study is among the first to compare indices of systemic inflammation (SIR and SIRI) with CA 125 in the differential diagnosis of benign formations, BOTs and OC, offering new diagnostic perspectives. Moreover, the multicenter design ensures that the results are more generalizable than single-center studies. We minimized potential selection biases by including multiple institutions and increased our findings’ external validity. Additionally, the use of strict inclusion criteria allowed us to analyze a well-defined and homogeneous population, ensuring that the observed differences were genuinely related to ovarian tumor biology rather than confounding inflammatory conditions. However, it has limitations that cannot be avoided. One weakness is related to the very strictness of the inclusion criteria, which limits applicability in everyday clinical practice.

Additionally, while the sample size was statistically adequate, larger-scale studies are needed to confirm the reliability of these inflammatory indices across diverse patient populations. Another important consideration is that this study was conducted exclusively in Italy, which may limit the applicability of the findings to other geographic regions. Genetic, environmental, and healthcare system differences could influence inflammatory responses and diagnostic performance. The generalizability of the results to other geographic regions and healthcare systems should be considered with caution. Additionally, subgroup analyses may prove that BMI and comorbidities, such as metabolic syndrome and autoimmune disorders, may hypothetically impact systemic inflammatory indices and their diagnostic accuracy. Age and BMI were initially included as covariates in exploratory multivariable logistic regression models to control for potential confounding effects. However, these variables were not retained in the final models, as their inclusion did not improve diagnostic performance.

Similarly, transient and subclinical proinflammatory states can also potentially influence the results obtained. Further research is needed to determine whether these variables affect the robustness of SIRI and SIR as diagnostic tools in broader clinical settings, with multicenter collaborations that ensure maximum variability in the study population. Furthermore, our study focused on short-term diagnostic accuracy without evaluating the prognostic value of SIR and SIRI over time. A longitudinal follow-up would be necessary to determine whether these indices can also serve as prognostic markers for disease progression or treatment response.

Lastly, despite excluding patients with known chronic inflammatory diseases, transient inflammatory conditions (such as subclinical infections or physiological stress) could still have influenced SIR and SIRI values, introducing a potential confounding factor.

### 4.5. Future Perspectives

The results obtained from our study would benefit from validation with cohorts of patients more varied in ethnicity and underlying clinical conditions. In addition, the highlighted laboratory data could be explored in combination with other imaging data, such as MRI or data related to organ function, to expand the perspectives with which to approach the differential diagnosis of adnexal neoformations. In addition, the advancement of technologies and artificial intelligence could provide an opportunity to design synthesis studies that encapsulate all the individual pieces useful for improving diagnostic power.

## 5. Conclusions

Our study highlights the potential diagnostic role of systemic inflammatory indices (SIR and SIRI) in differentiating benign from non-benign adnexal masses (BOTs and OC). Compared to CA 125, these markers demonstrated higher specificity and diagnostic accuracy, making them valuable tools in preoperative assessment. The ability of SIRI to outperform CA 125 in distinguishing benign from non-benign conditions suggests that systemic inflammation plays a key role in tumor progression. Furthermore, SIR and SIRI are derived from a simple complete blood count (CBC), so they offer a cost-effective and widely accessible alternative to traditional biomarkers. However, our study was conducted under strict inclusion criteria, limiting its generalizability, and more extensive multicenter studies are required to validate these findings. Future research should explore the integration of inflammatory indices with imaging techniques to improve diagnostic models further. Only after further results have been obtained could these indices be useful in practice.

## Figures and Tables

**Figure 1 jpm-15-00426-f001:**
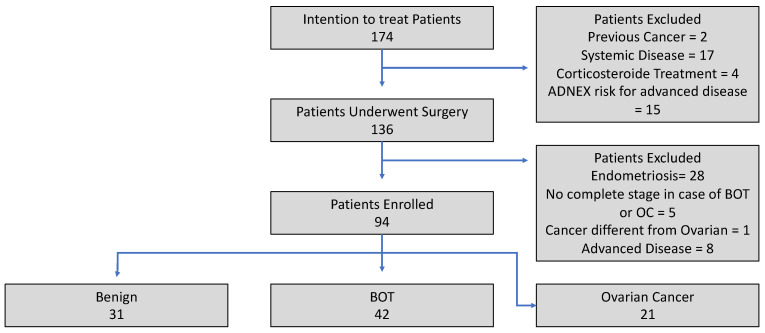
Enrollment flowchart.

**Figure 2 jpm-15-00426-f002:**
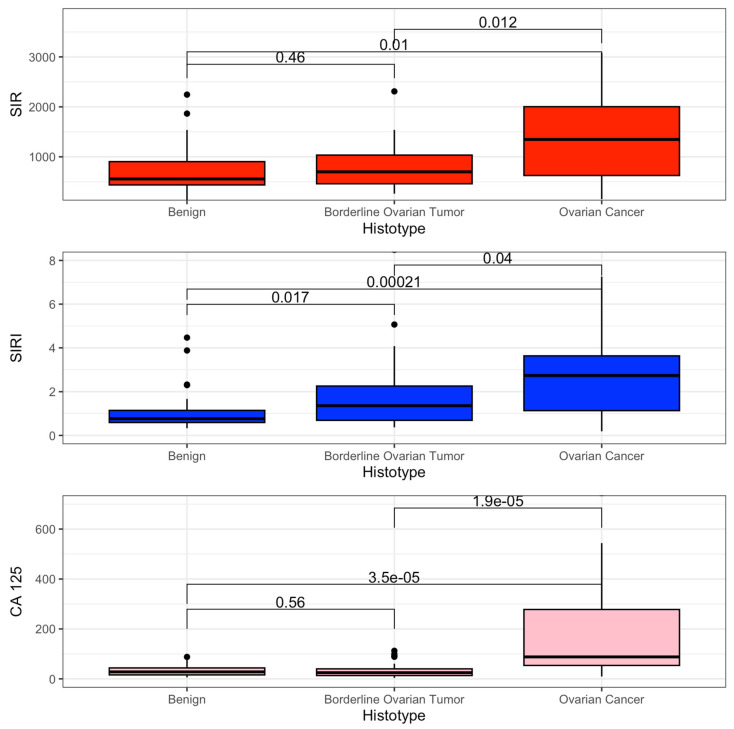
Boxplot SIR, SIRI, and CA 125.

**Figure 3 jpm-15-00426-f003:**
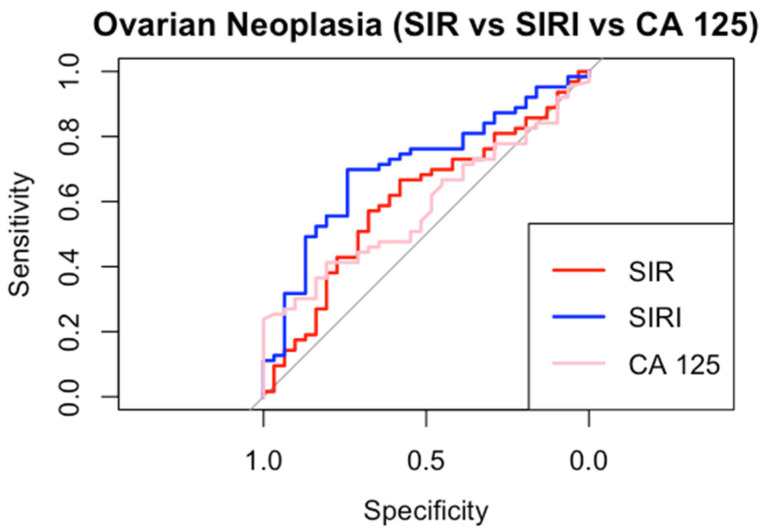
ROC curves.

**Table 1 jpm-15-00426-t001:** Patient characteristics.

Characteristic	Benign, N = 31 ^1^	Borderline Ovarian Tumor, N = 42 ^1^	Ovarian Cancer, N = 21 ^1^	*p*-Value ^2^
Age	48 (35, 58)	51 (38, 64)	56 (49, 63)	0.094
BMI	24.2 (22.0, 27.5)	23.0 (19.4, 27.6)	25.7 (23.0, 29.3)	0.169
Menopause	11 (35%)	20 (48%)	14 (67%)	0.086
Histology				
Serous Cystadenoma	16 (52%)	-	-	
Mucinous Cystadenoma	3 (9.7%)	-	-	
Mature Cystic Teratoma	9 (29%)	-	-	
Ovarian Fibroma	1 (3.2%)	-	-	
Struma Ovarii	2 (6.5%)	-	-	
BOT Endometrioid	-	1 (2.4%)	-	
Mucinous BOT	-	19 (45%)	-	
Serous BOT	-	22 (52%)	-	
HGSOC	-	-	9 (43%)	
LGSOC	-	-	4 (19%)	
Mucinous Expansive	-	-	2 (9.5%)	
Endometrioid	-	-	1 (4.8%)	
Clear Cell	-	-	3 (14%)	
Stromal	-	-	1 (4.8%)	
CA 125	28 (16, 44)	25 (13, 40)	88 (54, 278)	**<0.001**
Neutrophils	3.96 (3.38, 5.13)	4.48 (3.49, 6.19)	6.63 (4.60, 7.20)	**0.014**
Lymphocytes	1.78 (1.46, 2.06)	1.68 (1.24, 2.07)	1.66 (1.28, 2.16)	0.491
Monocytes	0.34 (0.28, 0.44)	0.44 (0.35, 0.59)	0.60 (0.52, 0.70)	**<0.001**
Platelets	257 (214, 308)	234 (199, 276)	300 (262, 371)	**0.008**

^1^ Median (Q1–Q3); n (%), ^2^ Kruskal–Wallis rank sum test; Fisher’s exact test, BOT: borderline ovarian tumor, HGSOC: high-grade serous ovarian cancer, LGSOC: low-grade serous ovarian cancer.

**Table 2 jpm-15-00426-t002:** Inflammation outcomes.

Characteristic	Benign, N = 31 ^1^	Borderline Ovarian Tumor, N = 42 ^1^	Ovarian Cancer, N = 21 ^1^	*p*-Value ^2^
SIR	557 (437, 903)	699 (460, 1034)	1347 (626, 2004)	**0.016**
SIRI	0.76 (0.59, 1.15)	1.36 (0.69, 2.26)	2.73 (1.13, 3.63)	**<0.001**

^1^ Median (Q1–Q3), ^2^ Kruskal–Wallis rank sum test.

**Table 3 jpm-15-00426-t003:** Linear regression.

	SIR			SIRI			CA 125		
Characteristic	Beta	95% CI ^1^	*p*-Value	Beta	95% CI ^1^	*p*-Value	Beta	95% CI ^1^	*p*-Value
Ovarian Neoplasia	310	−216, 836	0.2	1.2	0.30, 2.1	**0.010**	111	−26, 249	0.11

^1^ CI = confidence interval.

**Table 4 jpm-15-00426-t004:** Predictive value of ovarian neoplasia.

Index	Cut Off	Sensitivity	Specificity	PPV	NPV
SIR ^1^	**710**	0.57 (0.44–0.69)	0.68 (0.49–0.83)	0.78 (0.64–0.89)	0.44 (0.29–0.59)
SIRI ^1^	**1**	0.70 (0.57–0.81)	0.74 (0.55–0.88)	0. 85 (0.72–0.93)	0.55 (0.39–0.70)
CA 125 ^2^	**39** Iu/dL	0.46 (0.33–0.59)	0.68 (0.49–0.83)	0.74 (0.58–0.87)	0.38 (0.25–0.52)

A 95% confidence interval, PPV: positive predictive value, NPV: negative predictive value, ^1^ calculated by Youden Index, ^2^ gold standard [28].

## Data Availability

Data regarding any of the subjects in this study has not been previously published. Data will be made available to the editors of the journal pre- and/or post-publication for review or query upon request. This study used the STROBE statement for observational studies [26]. All data and the methodological process for their calculation can be supplied under explicit request to the corresponding author and provided as an ‘R’ file.

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
