# Peer review of "Inflammatory Indices vs. CA 125 for the Diagnosis of Early Ovarian Cancer: Evidence from a Multicenter Prospective Italian Cohort"

_jpm, 2025, doi:10.3390/jpm15090426_

Round 1

Reviewer 1 Report

Comments and Suggestions for Authors

General remarks:
This prospective multicenter observational study investigates the diagnostic performance of two systemic inflammatory markers—Systemic Inflammation Response Index (SIRI) and Systemic Inflammatory Response (SIR)—compared to the widely used tumor biomarker CA 125 in women with suspicious ovarian masses. A total of 94 women with adnexal masses were enrolled, including 31 with benign tumors, 42 with borderline ovarian tumors (BOT), and 21 with ovarian cancer (OC). The study found that SIRI had a higher diagnostic accuracy (AUC = 0.71) than CA 125 (AUC = 0.59), with statistically significant associations between elevated SIRI values and non-benign tumor histology. The authors conclude that inflammatory indices, particularly SIRI, may serve as cost-effective, accessible tools for the preoperative stratification of ovarian masses.

-Introduction:

This section provides an appropriate overview of the diagnostic challenges in ovarian cancer, particularly the limitations of CA 125 and imaging-based tools like the ADNEX model. The rationale for investigating systemic inflammatory indices is compelling, grounded in emerging oncologic literature. However, the transition to these indices would be stronger if more emphasis were placed on how their biological relevance directly supports preoperative risk stratification in ovarian tumors.

Additionally, the background on limitations of CA 125 could benefit from more direct comparisons to other candidate biomarkers evaluated in the literature.

The study objective is stated clearly and concisely. Nonetheless, this section would be strengthened by formally dividing it into primary and secondary objectives, which would help structure the subsequent methods and results sections. The emphasis on comparing inflammatory indices with CA 125 is appropriate, although the hypothesis could be rephrased in a more neutral and academic tone.

- Discussion:
This section is broadly well-written, with the authors providing biologically plausible mechanisms linking systemic inflammation and tumor progression. They interpret the findings in light of current evidence, emphasizing the inflammatory milieu of malignant ovarian disease. However, at times the tone becomes overly informal or subjective. The sentence:  “our desire to differentiate what is benign...”, could be reframed as: “The need to accurately distinguish benign from non-benign adnexal masses...”

Also, the claim that SIRI outperforms CA 125 should be tempered, given that an AUC of 0.71 represents only moderate diagnostic utility. The clinical implications are interesting and relevant, particularly the potential cost-effectiveness of SIRI/SIR derived from routine blood tests, but remain speculative without a detailed economic analysis or validation in larger, more heterogeneous cohorts. This issue should be added as a remark.

In the comparison with the existing literature, the authors draw appropriately from recent studies showing that systemic inflammatory markers are elevated in early-stage malignancies. However, some cited studies, especially those on COVID-19 inflammatory markers, are tangential and should be either removed or better contextualized. More robust comparisons with alternative biomarkers (e.g., HE4, ROMA score) would strengthen the discussion.

The limitations section is candid and acknowledges the impact of stringent inclusion criteria, modest sample size, and the exclusive Italian cohort. Additional limitations should include the potential influence of subclinical or transient inflammation, which could not be completely ruled out, and the lack of prognostic analysis over time. Also, it should more clearly state that further multicenter validation is needed before clinical adoption.

Several typographical errors are present (e.g., “Menopouse,”, “Platets”, “Monocytis” and “Lymphocytis” in Table 1). 

Author Response

1- “However, the transition to these indices would be stronger if more emphasis were placed on how their biological relevance directly supports preoperative risk stratification in ovarian tumors.

Additionally, the background on limitations of CA 125 could benefit from more direct comparisons to other candidate biomarkers evaluated in the literature.”

1- Thank you for your comments, we found them indispensable to improve the quality of our study. Regarding what you underlined, we emphasized in the introduction the rationale behind the use of indices of inflammation diagnostically as given here: “In addition, they can potentially reflect the individual's response to tu-moral progression. it is likely that upon the tumor tissue's acquisition of infiltrative capacity, it comes in contact with the patient's immune system, resulting in its response, which can potentially be measured.”

------------

2- “The study objective is stated clearly and concisely. Nonetheless, this section would be strengthened by formally dividing it into primary and secondary objectives, which would help structure the subsequent methods and results sections. “

2- We have divided the goal section according to your comment. Thank you

------------

3- “The emphasis on comparing inflammatory indices with CA 125 is appropriate, although the hypothesis could be rephrased in a more neutral and academic tone.”

3- Abbiamo riformulato l’ipotesi nulla mantenendo un tono maggiormente accademico: (This study aims to evaluate whether inflammatory indices (SIRI and SIR) are elevated in patients with ovarian cancer (OC) and borderline ovarian tumors (BOT), and whether their combination improves diagnostic performance compared to CA 125 alone)

------------

4- The sentence:  “our desire to differentiate what is benign...”, could be reframed as: “The need to accurately distinguish benign from non-benign adnexal masses...”

4- Thank you for the suggestion. We have made this change

------------

5- “Also, the claim that SIRI outperforms CA 125 should be tempered, given that an AUC of 0.71 represents only moderate diagnostic utility.”

5- Thank you, we have moderated the tone and enthusiasm in the discussion session, bringing back to the reader more interpretive caution

------------

6-  “In the comparison with the existing literature, the authors draw appropriately from recent studies showing that systemic inflammatory markers are elevated in early-stage malignancies. However, some cited studies, especially those on COVID-19 inflammatory markers, are tangential and should be either removed or better contextualized. More robust comparisons with alternative biomarkers (e.g., HE4, ROMA score) would strengthen the discussion.”

6- We agree that some studies deserve better context, so we have expanded the discussion section

------------

7- “The limitations section is candid and acknowledges the impact of stringent inclusion criteria, modest sample size, and the exclusive Italian cohort. Additional limitations should include the potential influence of subclinical or transient inflammation, which could not be completely ruled out, and the lack of prognostic analysis over time. Also, it should more clearly state that further multicenter validation is needed before clinical adoption.”

7- Thank you, we specified in the limitations the uncertainty of finding transient pro-inflammatory conditions, which may have played a confounding role. In addition, we added that future validations should be multicenter in nature

------------

8- “Several typographical errors are present (e.g., “Menopouse,”, “Platets”, “Monocytis” and “Lymphocytis” in Table 1).”

8- We conducted a rigorous review of English and spelling. Thank you.

------------

Thank you for taking the time to evaluate our study. In addition to this detailed response, you can find attached an updated version of the manuscript with all changes highlighted

Reviewer 2 Report

Comments and Suggestions for Authors

The article explores the diagnostic performance about ovarian cancer of two indices: the Systemic Inflammatory Response (SIR), obtained by multiplying the neutrophil count by the platelet count and dividing by the lymphocyte count, and the Systemic Inflammatory Response Index (SIRI), determined by multiplying monocytes by platelets and dividing by lymphocytes. The diagnostic performance is compared at Ca125, a routine marker. This article provides some interesting insights into the use of biological data already available in standard blood counts.

General comments

The subject of this study is a frequent clinical problem, propose new tools for the pre-operative diagnosis of adnexal masses between benign masses, BOT and ovarian cancers. The two proposed indices do not entail any additional cost, since they are calculated on the basis of blood counts, which are routinely available.

The methodology is clear and well explained. The English language is understandable. The bibliography is adapted.

  • Specific comments 

Introduction

The introduction clearly sets out the context and objectives of the study.

Methods

The choice of methodology was to exclude pre-operative patients, for example with chronic inflammatory diseases, a choice explained by the authors and with an impact on the generalizability of the results also addressed in the discussion.

On the other hand, it would be interesting to explain/discuss the reasons for intraoperative exclusions. It is these cases of exclusion in particular that compromise the generalizability, of results.

Results

The results are clear and the figures easy to understand. Figure 2 could show the “p” between each of the 3 box plots.

Discussion

The discussion is clear and divides in well-balanced paragraphs. One or two sentences could comment on the results between BOT and benign mass.

Conclusion

The conclusion summarizes the main findings. The prospect of testing these two indices in less restrictive clinical contexts is a fair one. The last sentence should be modified: only after further results have been obtained could these indices be useful in practice.

Supplementary data: supplementary data do not add new data

Author Response

Thank you for your words of esteem and appreciation of our manuscript and for your attention to it.

Below we try to respond to your valuable comments, which we have found helpful in improving our work.

1- “On the other hand, it would be interesting to explain/discuss the reasons for intraoperative exclusions. 

1- Thank you for your observation. We reported that the intraoperative exclusion was related to the diagnosis of endometriosis, and therefore pro-inflammatory condition, found in the operating room and not previously diagnosed

---

2- “Figure 2 could show the “p” between each of the 3 box plots.”

2- We have reedited Figure 2 so that it includes the p of confront among all categories

---

3- “One or two sentences could comment on the results between BOT and benign mass.”

3- Thank you, we have added a consideration regarding the fact that even between BOT and benign SIRI was found to be statistically different “the mechanisms underlying malignant transformation appear to drive distinct immune responses in benign neoplasms, BOT, and OC. In line with this observation, SIRI was also shown to differ significantly between benign neoformations and BOT.”

---

4- “The last sentence should be modified: only after further results have been obtained could these indices be useful in practice.”

4- We changed the last sentence in accordance with your observation

----

In general, thank you for your careful comments that helped to improve our manuscript. We hope we have cleared up the doubts that were raised. Please also be advised that we have made an additional check of the grammar and spelling and that all changes made are highlighted in the manuscript uploaded in response to these revisions

Reviewer 3 Report

Comments and Suggestions for Authors

This is a well-designed prospective multicenter study exploring the diagnostic value of SIRI and SIR versus CA 125 in ovarian masses. The manuscript addresses a relevant clinical issue and presents statistically supported conclusions. However, several improvements are necessary before publication.

  1. Chapter and subchapter titles should be more descriptive and informative.
  2. Please revise keywords using MeSH terms for better indexing and discoverability.
  3. Add a clear Limitations section. Discuss sample size, strict exclusion criteria, lack of longitudinal data, and geographic scope.

  4. Include a Future Perspectives paragraph. Suggest validation in larger, more diverse cohorts and integration with imaging models.

  5. Clarify what current knowledge is lacking in this area and how this study fills the gap.

  6. Many paragraphs lack citations to support claims—especially in the discussion. Each main conclusion should be referenced.

  7. Add more up-to-date references (2023–2024) from high-impact journals to strengthen the scientific basis.

  8. Cite key papers on diagnostic biomarkers, cost-effectiveness of CBC-derived indices, and recent advances in ovarian cancer diagnostics.

  9. Expand the “Laboratory” section slightly to clarify the quality control protocols across centers.

  10. Provide reference or rationale for cut-off values and Youden index usage.

11. The regression analysis should better clarify covariate handling (e.g., age, BMI).

12. Consider using multinomial logistic regression instead of linear regression for categorical outcomes (Benign vs BOT vs OC).

13. Use consistent terminology (e.g., “non-benign tumors” vs “BOT and OC”).

14. A graphical abstract would improve visual summary of findings.

15. Consider to citing https://doi.org/10.3389/fonc.2022.954008

Author Response

Thank you for your words of esteem and appreciation of our manuscript and for your attention to it.

Below, we try to respond to your valuable comments, which we have found helpful in improving our work.

1-  Chapter and subchapter titles should be more descriptive and informative.

1- We changed the title in: Inflammatory Indices vs CA 125 for the Diagnosis of Early Ovarian Cancer: Evidence from a Multicenter Prospective Italian Cohort

--

2- Please revise keywords using MeSH terms for better indexing and discoverability.

2- We changed the Keywords with equivalent mesh terms

---

3- Add a clear Limitations section. Discuss sample size, strict exclusion criteria, lack of longitudinal data, and geographic scope.

3- Thank you. In the discussion section in the section on limitations, you will find in a discursive way all the limitations that you have pointed out to us

---

4- Include a Future Perspectives paragraph. Suggest validation in larger, more diverse cohorts and integration with imaging models.

4- We added a discussion paragraph on Future prospects in which we contextualized the possibility of using computational studies and fusion with other imaging data in order to be able to improve the applicability of the results.

---

5- Clarify what current knowledge is lacking in this area and how this study fills the gap.

5- In the introduction section, we expanded on what may be the benefits that can be derived from our study

---

6- Many paragraphs lack citations to support claims—especially in the discussion. Each main conclusion should be referenced.

6- Thank you for your comment. We fully acknowledge the importance of supporting each statement with appropriate references, particularly in the discussion section. We apologize if some parts of the manuscript appeared assertive or insufficiently referenced. However, we were not entirely able to identify the specific passages you are referring to. We would be happy to provide additional relevant citations, and we kindly ask if you could indicate the specific points you consider lacking in references so that we can address them appropriately and effectively. We remain fully available to implement any necessary revisions.

---

7-Add more up-to-date references (2023–2024) from high-impact journals to strengthen the scientific basis.

7- We appreciate your suggestion regarding the inclusion of more recent references. To date, a meta-analysis on the Platets/lymphocytes ratio in ovarian carcinomas has been published on the topic since the previous citations, which is now included in our paper

---

8- Cite key papers on diagnostic biomarkerscost-effectiveness of CBC-derived indices, and recent advances in ovarian cancer diagnostics.

8- Thank you, we have strengthened the scientific contextualization by reporting two additional papers on the costs and benefits of using systemic inflammatory indeces

---

9- Expand the “Laboratory” section slightly to clarify the quality control protocols across centers.

9- Thank you for your valuable suggestion. We have revised the "Laboratory" section to include a more detailed description of the quality control procedures adopted across participating centers. Specifically, we have clarified that all centers followed standardized pre-analytical and analytical protocols, with regular internal and external quality assessments in accordance with national guidelines. This ensures consistency and reliability of the hematological and biochemical parameters used in the analysis.

---

10- Provide reference or rationale for cut-off values and Youden index usage.

10- Thank you for pointing this out. The cut-off values used in our analysis were determined based on the Youden index, which is a widely accepted method for optimizing diagnostic test performance by maximizing the sum of sensitivity and specificity. We have now added a reference to support the use of this approach (Youden, W.J. (1950). Index for rating diagnostic tests. Cancer, 3(1), 32–35). Additionally, we have clarified in the Methods section that cut-off values were derived from our ROC curve analyses, rather than being based on pre-established thresholds.

---

11- The regression analysis should better clarify covariate handling (e.g., age, BMI).

11- Thank you for your observation. We agree that the handling of covariates requires clearer explanation. As requested, we have revised the Methods section to specify that variables such as age and BMI were initially considered as potential confounders and included in preliminary multivariable logistic regression models. However, the final models presented in the manuscript were selected based on diagnostic performance (AUC, sensitivity, specificity) and excluded these variables, as their inclusion did not improve the accuracy of the models. We have updated the text to make this rationale explicit and added a sentence clarifying this in the Discussion sections.

---

  1. Consider using multinomial logistic regressioninstead of linear regression for categorical outcomes (Benign vs BOT vs OC).

12- Thank you for your suggestion. We agree that multinomial logistic regression would be appropriate for modeling a three-category outcome such as Benign vs BOT vs OC. However, in our study, the primary aim was to evaluate the diagnostic performance of inflammatory indices in discriminating between benign and malignant ovarian conditions. Therefore, we intentionally dichotomized the outcome to align with our clinical research question and to facilitate comparison with standard diagnostic tools such as CA 125. This binary approach also allowed us to calculate performance metrics such as sensitivity, specificity, and AUC, which are not directly applicable in multinomial models. Nonetheless, we acknowledge the potential of multinomial logistic models in future analyses focused on more granular diagnostic differentiation.

---

  1. Use consistent terminology(e.g., “non-benign tumors” vs “BOT and OC”).

13- Thank you, we have standardized the terminology used

---

  1. graphical abstractwould improve visual summary of findings.

14- Since it is not mandatory, and since there are costs for its production related to the involvement of professional figures, in the absence of funds, we preferred not to attach a graphical abstract

---

15- Consider to citing https://doi.org/10.3389/fonc.2022.954008

15- The recommended study, entitled "Clinical and molecular evaluation of patients with ovarian cancer in the context of drug resistance to chemotherapy," is undoubtedly of very high scientific standing. However, it talks about changes in the genetic expression of patients undergoing chemotherapy who have developed drug resistance. That topic is extremely far removed from those covered in our article. Although it deals with changes in CA 125 expression, we struggle to find a logical connection that could justify its citation. Was there perhaps a mistake? Were you referring to another article?

In general, thank you for your careful comments that helped to improve our manuscript. We hope we have cleared up the doubts that were raised. Please also be advised that we have made an additional check of the grammar and spelling and that all changes made are highlighted in the manuscript uploaded in response to these revisions

Round 2

Reviewer 1 Report

Comments and Suggestions for Authors

Corrections had significantly improved the manuscript.

Author Response

Thank You

Reviewer 3 Report

Comments and Suggestions for Authors
  1. Authors acknowledge the point and ask for specific locations. They did add multiple high-quality citations throughout. While referencing has been improved, some conclusions (e.g., clinical utility of inflammatory indices) could benefit from additional supporting citations. Still, their proactive request for specificity is reasonable.
  2. Consider to citing https://doi.org/10.3389/fonc.2022.954008 The citation of the article by Opławski et al. (2022) is well justified, as it provides compelling evidence for the prognostic relevance of both molecular and inflammatory biomarkers in the context of cancer treatment response. This directly supports and reinforces the conclusions presented in the reviewed manuscript. Incorporating this study strengthens the rationale for the potential clinical utility of the analyzed inflammatory indices and highlights the growing importance of biomarker-based approaches in contemporary oncology.

Author Response

We thank you for the valuable suggestion and fully agree that supporting the conclusions with further evidence would enhance the manuscript's scientific rigor. In response, we have included a citation to the study by Opławski et al. (Front. Oncol. 2022; https://doi.org/10.3389/fonc.2022.954008). The citation has been added in the “Discussion – Clinical Implication” section